# Magnitude and associated factors of sexual violence among female housemaids attending night school in Bahir Dar City, Northwest Ethiopia: Institution-Based Cross-Sectional Study, 2022

**Seble Asheber Gebremariam[1], Mulugeta Yaze Gebey[1], Zemenu Shiferaw Yadita[2], Yibeltal Alemu Bekele[2] ***

**1** Bahir Dar University, Bahir Dar, Ethiopia, **2** Department of Reproductive Health and Population Studies, School of Public Health, College of Medicine and Health Sciences, Bahir Dar University, Bahir Dar, Ethiopia

* yibeltalalemu6@gmail.com

## Abstract

### Background

Sexual violence is a major public health problem that affects the health and well-being of millions of young people. Housemaids are the most vulnerable group to sexual violence due to their nature of work. It leads to various physical, mental, sexual, and reproductive health problems, especially in our country's context. However, they have limited information regarding the magnitude and the factors associated with sexual violence among housemaids. As a result, the purpose of this study was to determine the magnitude and risk factors for sexual violence among female housemaids attending night school in Bahir Dar, Northwest Ethiopia, in 2022.

### Method

An institution-based cross-sectional study design was conducted among 340 housemaids attending night school in Bahir Dar city from May 15 to June 20, 2022. Participants were selected using simple random sampling through computer-generated techniques. An interviewer-administered, structured questionnaire was used. Data were entered, coded, and cleaned using EPI Data version 4.6.0.2, and exported to SPSS version 26 for further analysis. Both bivariable and multivariable logistic regression were done to identify factors associated with sexual violence. P-value and 95% confidence interval were used to declare the statistical association.

### Result

The magnitude of sexual violence after being a housemaid was 30.3% with a 95% confidence interval (25.3–35.38). Ever had sex [Adjusted Odds Ratio(AOR) = 4.67; 95%; Confidence Interval(CI) (2.60, 8.39)], no discussion of sexual and reproductive issues [Adjusted

**Data Availability Statement:** All necessary data were included in the paper. Raw data is available from the corresponding author upon reasonable request.

**Funding:** The author(s) received no specific funding for this work.

**Competing interests:** The authors have declared that no competing interests exist.

Odds Ratio(AOR) = 2.32; 95%; Confidence Interval(CI) (1.29, 4.16)], poor social support [Adjusted Odds Ratio(AOR) = 2.69; 95%; Confidence Interval(CI) (1.32, 5.52)], were identified as factors associated with sexual violence among housemaids. Similarly, academic performance [Adjusted Odds Ratio (AOR) = 0.96; 95%; Confidence Interval (CI) (0.93, 0.99)], and distance to reach school [Adjusted Odds Ratio (AOR) = 2.04; 95%; Confidence Interval (CI) (1.19, 3.48)] were identified as factors associated with sexual violence among housemaids.

## Conclusion

This study identified that the magnitude of sexual violence among housemaids was high. Housemaids who ever had sex, no discussion of sexual and reproductive health (SRH) with anyone, poor social support, poor academic performance, and distance to reach school were factors associated with sexual violence. Therefore, creating a sexual and reproductive health (SRH) discussion session for housemaids is important for securing their sexual rights.

## Background

Violence is the intentional use of physical force or power, whether threatened or actual, against another person, which results in or has a high likelihood of resulting in injury, death, or psychological harm. While the majority of violence survivors are girls and women, boys and men can also be victims [1]. Sexual violence (SV) is defined as any sexual act or attempt carried out without the individual's consent, in any setting. It is common among people living in areas of insecurity, social disparity, cultural influences, lack of power, poverty, behavioural disturbances, and mental instability. Common forms of sexual violence include threats of rape, attempted rape, completed rape, sexual harassment, and unwanted sexual contact [1–3]. SV causes serious short- and long-term health problems for victims, including physical injuries, mental health issues, sexual and reproductive problems, unwanted pregnancy, unsafe abortion, and school dropouts [4,5]. Minority and marginalized women, who are among the least protected groups under national and international law, are particularly vulnerable to all forms of violence. Housemaids, being one of the most marginalized groups due to the nature of their work, economic status, and educational level, are at a high risk of sexual violence [6–9].

There are 75.6 million domestic workers worldwide, with 76% being female. Most domestic employees are directly employed by households, with over half (55%) coming from Latin America, the Caribbean, and East and Southeast Asia. Nearly 90% of domestic workers globally are employed informally and lack sufficient social or labor protections, making them more vulnerable to various types of violence, including physical abuse, intimidation, threats, bullying, sexual assault, and harassment. Despite significant global and national efforts to address violence, the burden of sexual violence among housemaids remains under-researched. Worldwide, 6% of women report being victims of non-partner sexual violence, with a similar prevalence in sub-Saharan nations and 10% of women in Ethiopia experiencing sexual violence at some point in their lives. All women and girls face certain types of violence, but indigenous and minority domestic workers endure a double tragedy. In Ethiopia, an estimated 248,600 women provided domestic work in 2018, with reports indicating that housemaids frequently experience forced sex and other forms of violence [10–20].

While research in Sub-Saharan Africa shows a range of 8.6% to 20.5%. In Ethiopia, the prevalence of sexual violence against female housemaids ranges from 27.8% to 72% in regions such as the Geode Zone, Harar, Addis Ababa, Debre Tabor, and Bahir Dar.

Global studies indicate that the prevalence of sexual violence among female housemaids ranges from 14% to 29% [21–25], while research in Sub-Saharan Africa shows a range of 8.6% to 20.5% [26–28]. In Ethiopia, the prevalence of sexual violence against female housemaids ranges from 27.8% to 72% in regions such as the Geode Zone, Harar, Addis Ababa, Debretabor, and Bahir Dar [29–34]. Identified risk factors for sexual violence include age, childhood residence, educational status, monthly salary, work experience, school performance, discussion of sexual and reproductive issues, alcohol use, family support, number of family members, social support, number of sexual partners, and distance from school [29,31,33–38].

Sexual violence holds a prominent position on the global agenda, being integrated into various international and national objectives. For instance, the Sustainable Development Goals aim to eradicate all forms of violence (Goal 5) and enhance educational achievements (Goal 4) by 2030. Similarly, Ethiopia's Growth and Transformation Plan II prioritizes addressing violence as a key focus area. However, the school health programs have not adequately addressed sexual violence and other forms of violence [39,40]. Presently, tackling sexual violence is crucial for achieving these international and national targets and enhancing the overall well-being of victims. However, there is a lack of comprehensive data on the current prevalence and associated factors, hindering the design of effective interventions. Hence, the objective of this study was to determine the prevalence of sexual violence and identify its determinant factors among female housemaids enrolled in night school in Bahir Dar, Amhara regional state, Northwest Ethiopia, 2022.

## Methods

### Study area, design and population

This study was conducted in Bahir Dar, the capital city of the Amhara region, which is around 565 km away from Addis Ababa, the capital city of Ethiopia. According to Bahir Dar's educational department, there are 93 schools in the city, of which 52 (41 primary and 11 secondary schools) are governmental and 41 are private. Sixteen schools provide an evening package, 15 of which are governmental and one non-governmental. A total of 6282 students enrolled in the evening program in the 2013 academic year, and 3712 of them were females [41]. An institution-based cross-sectional was conducted from May to June 2022. All female housemaids who were attending evening school in Bahir Dar city were the source population, while selected female housemaids who enrolled in the evening program in 2022/23 academic year were the study population.

### Sampling size determination and procedure

The sample size was calculated using the single population proportion formula by considering the following assumptions: the prevalence of sexual violence among housemaids in Debre Tabor Town was 27.8% [33], with a 95% confidence level, and a 5% margin of error. After adding the 10% non-response rate, a total of 340 housemaids participated in this study.

This study included all sixteen schools, both private and government, that offer evening programs. Initially, a preliminary census was conducted at each school to identify students working as housemaids by recording their school names, job types, grade levels, class numbers, and sections. Each eligible respondent was then assigned a unique code for identification. Participants were selected using a simple random sampling technique, utilizing computer-generated methods.

## Operational definition

**Sexual violence.** A WHO standard tool was used to assess sexual violence. Each housemaid was asked four independent Yes/No questions: Has your husband ever physically forced you to have sex or engage in other sexual acts against your will? Has your current or any former partner ever physically forced you to have sexual intercourse when you did not want to? Have you ever had sexual intercourse because you were afraid of what your partner or any other partner might do? has your partner or any other partner ever forced you to do something sexual that you found degrading or humiliating? has a housemaid answered "yes" to any of these questions, she was considered to have experienced sexual violence [42].

**Social support.** Social support was measured using the Oslo 3 Social Support Scale, which consists of three questions with a total possible score of 14 points. Participants scoring 3–8 were categorized as having poor social support, those scoring 9–11 were categorized as having moderate social support, and those scoring 12–14 were categorized as having good social support [43].

**Discussion on sexual and reproductive health issues.** Housemaids who had conversations with anyone in the last 12 months about at least two sexual and reproductive health (SRH) topics, such as physical and psychological changes during puberty, sexual intercourse, condoms, STI/HIV/AIDS, unwanted pregnancy, and contraception [44].

## Data collection procedure and data quality control

A structured questionnaire was developed by reviewing different literature [29,31,33–36,38,45,46] and the WHO violence measurement tools [42], were used for data collection. It was first written in English, then translated into Amharic, and then back to English to ensure consistency. The tool consists of socio-demographic, individual, and behavioral-related characteristics. After developing the questionnaire, both content and face validity were assessed by experts and non-experts in the subject area, respectively. Two midwives with BSc degrees and two psychiatry nurses with BSc degrees were assigned as data collectors, with one supervisor holding a Master of Public Health (MPH). Two days of training were given for both data collectors and supervisors on the objective, the purpose, how to approach the participants, and the content of the tools. A pretest was conducted among 10% (34 participants) of the participants to check the consistency and appropriateness of the tool outside of the study area. The supervisor controls the overall data collection process throughout the data collection time. While the principal investigator controls the data collection process daily.

## Data processing and analysis

Data were cleaned, coded, and entered using Epi-Data 4.6.0.2, then exported to SPSS version 26 for further analysis. Descriptive statistics were computed for the selected variables. Binary logistic regression was employed to identify factors associated with sexual violence. Bi-variable logistic regression was employed to select the candidate variable for multivariable analysis. Those variables whose P-value was less than 0.25 entered into the multivariable logistic regression. The strength of the association was measured by the adjusted odds ratio, while the 95% confidence interval and p-value were used to declare a statistical association. Statistical significance was defined as P-values less than 0.05. The Hosmer-Lemeshow goodness test was fitted to check model fitness, which was found to be 0.65.

## Ethical consideration

Ethical clearance was obtained from the ethical review committee of Bahir Dar University, College of Medicine and Health Sciences, with the reference number IRB443/22. A support

letter was obtained from all the respective administrative bodies. Written consent was obtained from each participant. Prior to the interview, the data collectors endeavored to contact the parents of participants under 18 years old to obtain consent. Additionally, assent was obtained from these participants themselves. Those participants who experienced sexual violence were linked to community health agents. The data collectors ensured that participants understood the purpose, privacy, confidentiality, and protocols for storing and using the information. Furthermore, all participants were informed of their right to withdraw from the study at any point.

## Result

### Socio-demographic characteristics of the housemaids

A total of 323 housemaids participated in this study, with a response rate of 95%. Two hundred fifty-three (78.5%) of the participants were below 20 years of age, with a mean and standard deviation of 18 years (SD±2.28). Two hundred eighty-four (78.9%) of the participants attended primary education (grades 1–8). Two hundred sixty-eight (83%) of the respondents don't have a boyfriend. Three hundred four (94.1%) of the participants were on a permanent working agreement. Two hundred and eleven (65.3%) of their parents were alive, while thirty-eight (11.7%) of the participants' parents had passed away. Concerning family support, one hundred eighty (55.7%) reported poor family support. One hundred ninety-three (59.7%) of the houses had more than three family members (Table 1).

### Sexual history and behavioral characteristics of the housemaids

Among the total study participants, one hundred and five (32.5%) had a history of sexual intercourse. Among those who have a history of sex, forty-three (13.3%) participants were sexually active in the past 12 months. The mean age of first sexual intercourse was 16 years, with a standard deviation of (SD±2.3). Two hundred forty-four (75.5%) of the participants had consumed alcohol at some point in their lives. Two hundred eleven (65.3%) of the men in the household drank alcohol. Ten (3.1%) of participants have ever smoked a cigar (Table 2).

### Magnitude of sexual violence among housemaids

The magnitude of sexual violence after being a housemaid in this study was 30.3% with a 95% CI of (25.3, 35.38). Ninety (91.8%) of the victims experienced sexual violence in the previous twelve months. Forty-one (37.27%) of the perpetrators were employers, followed by 26 (23.6%) passengers, and 24 (21.82%) relatives of the housemaids. Sixty-five (66.33%) of the victims did not notify anyone, while sixteen (16.33%) did.

### Factors associated with sexual violence among housemaids

In bivariable analysis, ever having sex, discussion of sexual and reproductive health, male employer alcohol use, work experience, employer family size, social support, school performance, and distance to school were associated with sexual violence at a P-value less than 0.25. In multivariable analysis, ever having sex, discussion of SRH, social support, school performance, and school distance were found to be statistically associated with sexual violence among housemaids at a P-value less than 0.05.

The odds of sexual violence among housemaids who had ever had sex were 4.67 times higher than those with no history of sex [AOR: 4.67, CI: (2.60, 8.39)]. Housemaids who did not discuss sexual and reproductive issues had a higher risk of sexual violence than their counterparts [AOR: 2.32, CI: 1.29, 4.16]. The odds of sexual violence among housemaids who had

**Table 1. Socio-demographic characteristics of female housemaids attending night school in Bahir Dar Northwest Ethiopia2022. (N = 323).**

| Variable | Frequency | Percent (%) |
|---|---|---|
| **Age** | | |
| Less than 20 years | 257 | 79.5 |
| 20 and above | 66 | 20.5 |
| **Religion** | | |
| Orthodox | 307 | 95 |
| Muslim | 16 | 5 |
| **Grade level** | | |
| Primary (grades 1–8) | 284 | 87.9 |
| Secondary(grades 9–12) | 39 | 12.1 |
| **Experience as a housemaid** | 278 | 86.1 |
| ≤5 years | | |
| >5 years | 45 | 13.9 |
| **Work agreement** | | |
| Daily | 19 | 5.9 |
| Permanent | 304 | 94.1 |
| **Marital status** | | |
| Single | 307 | 95 |
| Divorced | 16 | 5 |
| **Have boyfriend** | | |
| Yes | 55 | 17 |
| No | 268 | 83 |
| **Salary** | | |
| 300–700 | 101 | 31.3 |
| 701–800 | 66 | 20.4 |
| 801–1000 | 128 | 39.6 |
| ≥1001 | 28 | 8.7 |
| **Parent alive** | | |
| Both are alive | 211 | 65.3 |
| One alive | 74 | 23 |
| Both are dead | 38 | 11.7 |
| **Family support** | | |
| Yes | 143 | 44.3 |
| No | 180 | 55.7 |
| **Family size of employer** | | |
| ≤3 | 130 | 40.2 |
| 4–6 | 193 | 59.8 |

poor social support were 2.69 times higher than those who had moderate social support (AOR: 2.69, CI: 1.32, 5.52). The odds of sexual violence among students who traveled more than 20 minutes to reach school were 2.04 times higher than those who traveled less than 20 minutes [AOR: 2.04, CI: (1.19, 3.48)] (Table 3).

## Discussion

These studies identify the magnitude of sexual violence after being a housemaid was 30.3% (95% CI: 25.3–35.38). The finding of the study is consistent with studies done in Addis Ababa 28.6%, Debre Tabor 27.8% respectively [31,33], Hong Kong 25.38% [47] and India 29% [25]. However, the finding of this study was lower than studies done in Bahir Dar 49.1% [34], Gedeo zone 60.2% [29], and Harare 72% [30]. This may be due to the socio-demographic variations of the study participants. All the participants in this study were attending formal

**Table 2. Sexual and behavioral characteristics of female housemaids attending night school in Bahir Dar Northwest Ethiopia2022 (N = 323).**

| Variable | Frequency | Percent (%) |
|---|---|---|
| **Ever had alcohol** | | |
| Yes | 244 | 75.5 |
| No | 79 | 24.5 |
| **Male household member who use alcohol** | | |
| Yes | 211 | 65.3 |
| No | 112 | 34.7 |
| **Ever chew chat** | | |
| Yes | 16 | 5 |
| No | 307 | 95 |
| **Ever smoke cigar** | | |
| Yes | 10 | 3.1 |
| No | 313 | 96.9 |
| **Discussion of SRH with partner/friends** | | |
| Yes | 134 | 41.5 |
| No | 189 | 58.5 |
| **Knowledge about sexual rights** | | |
| Not knowledgeable | 65 | 20.1 |
| Knowledgeable | 258 | 79.9 |
| **Social support** | | |
| Poor | 238 | 73.7 |
| Moderate | 85 | 26.3 |
| **Ever had sex** | | |
| Yes | 105 | 32.5 |
| No | 218 | 67.5 |
| **Age at first sex(n = 105)** | | |
| Less than 15 years | 16 | 15.24 |
| 15 to 19 | 86 | 81.9 |
| 20 to 24 | 3 | 2.86 |
| **Sex in the past 12 month** | | |
| Yes | 43 | 40.95 |
| No | 62 | 59.05 |

education, while study participants in Harar included housemaids who did not attend formal education, which increases vulnerability to sexual violence[48]. Similarly, the difference from the previous study conducted in Bahir Dar may be attributed to the characteristics of the participants. In the previous study, only 31% of participants lived with their employers, while 94% of participants in this study live with their employer [34].

On the other hand, the findings of this study were higher compared to a study done in Rwanda (8.67%) and Portugal (14%) [24,28]. This might be due to the tight implementation of laws against sexual violence compared to our study area. Portugal's government has recently improved the flexibility of its employment rules and made a number of adjustments to them. It serves as a good illustration of an effective and quick economic and labor market recovery that doesn't compromise employees' rights. Currently, the Portuguese labor Code provides more employer-friendly regulations regarding workforce organization by attempting to strike a fair balance between employees' rights and employers' freedom of management, which helps to minimize the impacts of violence among workers [49]. This implies that Ethiopian governments will use the lessons learned to create employer rules and regulations, including contractual agreements.

**Table 3. Bivariable and multi-variable regression analysis for factors associated with sexual violence among female housemaids attending night school in Bahir Dar Ethiopia 2022.** (N = 323).

| Variables | Sexual violence | | Odds ratio | | |
|---|---|---|---|---|---|
| | Yes (%) | No (%) | COR(95%CI) | AOR(95%CI) | P-value |
| **Male alcohol use** Yes | 74(22.9) | 137(42.4) | 1.98(1.16,3.37) | 1.69(0.93, 3.07) | 0.983 |
| No | 24(7.4) | 88(273) | 1 | 1 | |
| **Work experience** ≤5years | 79(24.5) | 199(61.5) | 0.54(0.29,1.04) | 0.73(0.34,1.58) | 0.06 |
| >5 years | 19(5.9) | 26(8.1) | 1 | 1 | |
| **Family size** 1–3 | 30(9.3) | 100(30.9) | 1 | 1 | |
| 4–6 | 65(20.1) | 117(36.2) | 1.85(1.11,3.07) | 1.49(0.8,2.7) | 0.207 |
| >6 | 3(0.9) | 8(2.6) | 1.25(0.31,5.01) | 1.25(0.26,6.09) | 0.53 |
| **Ever had sexual intercourse** Yes | 50(15.5) | 55(17) | 3.22(1.95,5.31) | 4.67(2.60, 8.39)** | 0.002 |
| No | 48(14.9) | 170(52.6) | 1 | 1 | |
| **Distance to reach school** <20 minutes | 47(14.6) | 149(46.1) | 1 | 1 | |
| 20–40 minutes | 51(15.8) | 76(23.6) | 2.13(1.31,3.45) | 2.04(1.19, 3.48)** | 0.015 |
| **Discussion of SRH** Yes | 28(8.7) | 106(32.8) | 1 | 1 | |
| No | 70(21.7) | 119(36.8) | 2.23(1.34,3.71) | 2.32(1.29, 4.16)** | 0.002 |
| **Social support** Poor | 84(26) | 154(47.7) | 2.77(1.47,5.20) | 2.69(1.32, 5.52)** | 0.001 |
| moderate | 14 (4.3) | 71(21.9) | 1 | 1 | |

Regarding the factors for sexual violence, history of sexual intercourse, discussion of sexual and reproductive issues, social support, school performance, and time taken to reach school were found to be independent predictors of sexual violence. Housemaids who have ever had sexual intercourse were more likely to be exposed to sexual violence than their counterparts were. Similarly, in Ethiopia and Jamaica, evidence suggests that the history of sexual activity does affect the occurrence of sexual violence [22,50]. This may be because individuals with a previous history of sexual intercourse often engage in risky behaviors that can increase their vulnerability to sexual violence [51].

This study identified that a housemaid who didn't discuss sexual and reproductive health issues increased the risk of sexual violence. This finding was consistent with studies conducted in Bahir Dar and Arbaminch [34,36]. The reason for this might be because information and discussion on sexual and reproductive matters will enable them to understand their sexual rights. This will further empower women to fight against gender-based violence, including sexual violence [52]. This implies that improving the discussion about the SRH issue may reduce the risk of sexual violence.

Housemaids who have had poor social support are more vulnerable to sexual violence compared with those who have moderate social support. This finding was consistent with studies conducted in Ethiopia and the USA [37,53]. This may be because housemaids may create a formal or informal relationship that helps them reduce the psychological and physiological consequences of stress and that provides a sense of belonging and security in the community [54].

The result of this study revealed that school performance was significantly associated with sexual violence [34,45]. Similarly, the time it takes to get to school is another factor in sexual violence. This is in congruence with other findings in Ethiopia that showed that the longer it takes to reach the school, the higher the risk of experiencing sexual violence [36].

## Conclusions

This study identified the magnitude of sexual violence among housemaid found to be high, compared to similar studies. Ever had sex, no discussion of sexual and reproductive health (SRH), poor social support, school performance, and time taken to reach school were found to be significant factors associated with housemaids' sexual violence. Therefore, developing a strategy to increase sexual and reproductive health information for night school students is critical for reducing sexual violence. In addition, providing psychosocial support for the victim is important to reduce the risk of adverse psychosocial effects.

## Acknowledgments

Firstly, the authors would like to express their gratitude to all participants who volunteered to take part in this study. We are also grateful to the Bahir Dar town education office staff, and schoolteachers for their invaluable support through the whole process and the supervisors and data collectors who have committed themselves throughout the study period.

## Author Contributions

**Conceptualization:** Seble Asheber Gebremariam, Mulugeta Yaze Gebey, Zemenu Shiferaw Yadita, Yibeltal Alemu Bekele.

**Formal analysis:** Zemenu Shiferaw Yadita.

**Investigation:** Seble Asheber Gebremariam.

**Methodology:** Seble Asheber Gebremariam, Mulugeta Yaze Gebey, Zemenu Shiferaw Yadita, Yibeltal Alemu Bekele.

**Visualization:** Mulugeta Yaze Gebey.

**Writing – original draft:** Seble Asheber Gebremariam, Mulugeta Yaze Gebey, Zemenu Shiferaw Yadita, Yibeltal Alemu Bekele.

**Writing – review & editing:** Seble Asheber Gebremariam, Mulugeta Yaze Gebey, Zemenu Shiferaw Yadita, Yibeltal Alemu Bekele.

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
