## [Decision Letter · Decision Letter 0]

12 Jan 2023

PONE-D-22-33222Magnitude and associated factors of sexual violence among female housemaids attending night school in Bahir Dar City, North West Ethiopia.PLOS ONE

Dear Dr. Bekele,

Thank you for submitting your manuscript to PLOS ONE. After careful consideration, we feel that it has merit but does not fully meet PLOS ONE’s publication criteria as it currently stands. Therefore, we invite you to submit a revised version of the manuscript that addresses the points raised during the review process.

We look forward to receiving your revised manuscript.

Kind regards,

Alemayehu Molla Wollie

Academic Editor

PLOS ONE

3. You indicated that you had ethical approval for your study. In your Methods section, please ensure you have also stated whether you obtained consent from parents or guardians of the minors included in the study or whether the research ethics committee or IRB specifically waived the need for their consent.

Reviewers' comments:

**Comments to the Author**

1. Is the manuscript technically sound, and do the data support the conclusions?

Reviewer #1: Yes

Reviewer #2: Partly

Reviewer #3: Yes

2. Has the statistical analysis been performed appropriately and rigorously? 

Reviewer #1: Yes

Reviewer #2: Yes

Reviewer #3: No

3. Have the authors made all data underlying the findings in their manuscript fully available?

Reviewer #1: Yes

Reviewer #2: Yes

Reviewer #3: Yes

4. Is the manuscript presented in an intelligible fashion and written in standard English?

Reviewer #1: No

Reviewer #2: Yes

Reviewer #3: No

5. Review Comments to the Author

Reviewer #1: I understand that this study will have an input for the existing knowledge if you revise the document thoroughly. Accordingly I have the following enlisted comments:-

1. The document is poorly written grammatically. I didn't understand some parts of the document. eg. under conclusion it says Therefore 'resolve this problem' ?? so what? this phrase lacks sense in your sentence.

2. The conclusion part in abstract is broader than and different from the one written in the main document. Please revise them

3. Some times you use abbreviation eg SRH which not internationally known and its meaning also varies in thedocument. so please make it similar either sexual and reproductive health or sexual and reproductive issue. In addition the term discussion can better be clear if you revise or add some term at a from like having discussion of sexual and reproductive health with family or having talk about sexual and reproductive health with partner pla..pla.. because the variable is vage.

4. one variable is also reported as time to school but under introduction the discussed factor is distance from school so please clearly justify under discussion or try to manage the variation.

5. under introduction the name of country and states are mixed and some reports lack evidence eg. It reports Brazil as a developed nation? considers texas as a country?

6. Under methods and materials section around line 8 please add study design and also write the study period in a similar way with the one found under Abstract section.

7. please specify the clear number of BSc Midwifes and Psychiatry nurse involved in your study?

8. I didnt like the way you wrote your result. You sometimes write in word and otherwise in figure it seems searching for way out from plagiarism. better to write both frequency and percentage in figures and specific to some findings from the table.

9. Summation of the percentages in the table is above 100. eg.for religion and marital status.

10. Under discussion please revise the name of countries or cities you used for discussion purpose and add more strong justifications.

Reviewer #2: The paper entitled Magnitude and associated factors of sexual violence among female housemaids attending night school in Bahir Dar City, North West Ethiopia addressed the public health concern that is sexual violence among the most venerable population communities in Ethiopia. The authors addressed very important points but the paper still needed to address the spelling, coherence, and scientific writing throughout the paper. This paper also failed to discuss, conclude and recommend based on the pertinent findings. Here I tried to address some of the comments that should be addressed.

Abstract; It is good but the structure and spelling errors; In the conclusion section of the abstract the factors should be written clearly for readers for example those who have poor school performance etc., the recommendation in the conclusion section is not specific and based on the pertinent finding.

Introduction; Grammar, punctuation, and the consequences of sexual violence are not briefed

Method; Separate the study area and study design separately, with a population (source and study population) as a subtitle. An operational definition should be clear (discussed briefly the items in short) and it should not be written all in one paragraph (one paragraph for one variable), specifying where the pretest was done. Data analysis section lines 6-8 is too long and unclear for readers (rewrite it scientifically). You do have an age of less than 20 participants means that there might be under 18 participants (checked SD±2.28), so how did you see the ethical concern of these participants?

Result: Table 1; do you mean male partner, the way that you measured the magnitude of sexual violence is not specifically briefed in the result section (you only discussed the general magnitude of sexual violence). Check and take some corrections on the odds of ……the way that you discussed the factors (AOR). Table 3; avoid COR, you should include the %value of each variable, and indicate with Astrix (**) the associated factor in the final model. The way how did you measure the discussion about SRH should be clearly shown in the operational definition; the variables that you use are somehow few; in the last section of table 3 what does the average score mean (Avoid it)?

Discussion; Line one said that These studies identify…..[correct it]……respectively (need correction) When you discuss the magnitude you should modify the grammar, the way that you discussed, and the way that you justify should be clear (It is good but it is not written scientifically). Discussion about factors must rewrite it as first your finding then the study which is consistent or inconsistent with your finding, reference, then possible justification (You should correct the factors section)

Conclusion; Is not addressed based on your pertinent finding and your objective

Check the reference some of your references do not exist in the online system

Reviewer #3: The grammar needs attention throughout the document, especially the introduction, results, and discussion sections. I do not agree to publish at this stage. The authors expect to revise exhaustively by incorporating all the above issues.

6. PLOS authors have the option to publish the peer review history of their article (what does this mean?). If published, this will include your full peer review and any attached files.

Reviewer #1: No

Reviewer #2: No

Reviewer #3: No

While revising your submission, please upload your figure files to the Preflight Analysis and Conversion Engine (PACE) digital diagnostic tool, https://pacev2.apexcovantage.com/. PACE helps ensure that figures meet PLOS requirements. To use PACE, you must first register as a user. Registration is free. Then, login and navigate to the UPLOAD tab, where you will find detailed instructions on how to use the tool. If you encounter any issues or have any questions when using PACE, please email PLOS at figures@plos.org. Please note that Supporting Information files do not need this step

---

## [Author Response · Author response to Decision Letter 0]

31 Jan 2023

Point by point response

 Reviewers comment 

 Authors response 

 Reviewer 1: 

1. The document is poorly written grammatically. I didn't understand some parts of the document. eg. under conclusion it says Therefore 'resolve this problem' ?? so what? this phrase lacks sense in your sentence. Thank you for your comment. To address the language issue, we consulted senior public health staff and language professors at my university. We also used online software, specifically Grammarly and scribens (to correct grammar and spelling), throughout the manuscript. 

In addition, Thank you for your comment. We accept rewrite this section as follows: “Therefore, creating a sexual and reproductive health (SRH) discussion session for housemaids is important for securing their sexual rights.” in the updated manuscript.

2. The conclusion part in abstract is broader than and different from the one written in the main document. Please revise them Thank you for your comment. We accept and update the conclusion section in the updated manuscript.

Line numbers 51 to 52

3. Sometimes you use abbreviation eg SRH which not internationally known and its meaning also varies in the document. so please make it similar either sexual and reproductive health or sexual and reproductive issue. In addition the term discussion can better be clear if you revise or add some term at a from like having discussion of sexual and reproductive health with family or having talk about sexual and reproductive health with partner pla..pla.. because the variable is vage. Thank you for your comment. We accept and define the abbreviation in the update manuscript. Discussion with sexual and reproductive health was measure whether the housemaids discuss any one( friends, partners, family members, employer and teachers) in the past twelve months and we revise it in the updated manuscript. 

4. One variable is also reported as time to school but under introduction the discussed factor is distance from school so please clearly justify under discussion or try to manage the variation. Thank you for your constructive comments. We measure distance from school with walking hours. Based on your comment we consistently used distance instead of time to school so we used distance to school in the updated manuscript. 

5. Under introduction the name of country and states are mixed and some reports lack evidence eg. It reports Brazil as a developed nation? Considers Texas as a country? Thank you for your comment. based on your comment we revised it as follows: “Studies conducted in the globe showed that the magnitude of sexual violence among female housemaids ranges from 14% to 29%” in the updated manuscript.

Line numbers 88 to 89 

6. Under methods and materials section around line 8 please add study design and also write the study period in a similar way with the one found under Abstract section. Thank you for your constructive comments. We amended all the comments in the updated manuscript.

Line numbers 88 to 89

7. Please specify the clear number of BSc Midwifes and Psychiatry nurse involved in your study? Thank you for your constructive comments. based on your comment we separately specify numbers of BSc Midwifes and Psychiatry nurse in the updated manuscript.

Line numbers 152 to 153

8. I didnt like the way you wrote your result. You sometimes write in word and otherwise in figure it seems searching for way out from plagiarism. better to write both frequency and percentage in figures and specific to some findings from the table. Thank you for your constructive comments. We amended all the comments in the updated manuscript.

9. Summation of the percentages in the table is above 100. eg.for religion and marital status. Thank you for your constructive comments. We amended all the comments in the updated manuscript.

Table 1

10. Under discussion please revise the name of countries or cities you used for discussion purpose and add more strong justifications. Thank you for the suggestion. Based on your comment, we revised the whole discussion section in the updated manuscript.

Line numbers 228 to 273

 Reviewer 2 

1. Abstract

It is good but the structure and spelling errors; 

 Thank you for your comment. To address the structural and spelling issue, we consulted language professors at my university and online software, specifically Grammarly and scribens (to correct grammar and spelling), throughout the manuscript.

 In the conclusion section of the abstract the factors should be written clearly for readers for example those who have poor school performance etc., 

 Thank you for the suggestion. Based on your comment, we clarify the variable in the updated manuscript.

Line number 54

 The recommendation in the conclusion section is not specific and based on the pertinent finding. Thank you for your constructive comments. We revised the recommendation section in the updated manuscript.

Line numbers 55 to 56

2. Introduction; Grammar, punctuation, and the consequences of sexual violence are not briefed Thank you for your comment. To address the language issue, we consulted senior public health staff and language professors at my university. We also used online software, specifically Grammarly and scribens (to correct grammar and spelling errors), throughout the manuscript.

In addition, we tried to highlight the consequences of sexual violence in the updated manuscript.

Line number 63 to 66

3. Method; 

Separate the study area and study design separately, with a population (source and study population) as a subtitle. Thank you for your constructive comments. We incorporate all your concern in the updated manuscript.

Line number 118 to 123

 An operational definition should be clear (discussed briefly the items in short) and it should not be written all in one paragraph (one paragraph for one variable), 

specifying where the pretest was done. 

 Thank you for your constructive comments. We incorporate all your concern in the updated manuscript.

Line number 136 to 149

The pretest was held in Debre Tabor Town, about 100 kilometers north of Bahir Dar. 

 Data analysis section lines 6-8 is too long and unclear for readers (rewrite it scientifically). Thank you for your suggestion. We revised the suggested section in the updated manuscript.

Line number 166 to 168

You do have an age of less than 20 participants means that there might be under 18 participants (checked SD±2.28), so how did you see the ethical concern of these participants? Yes, there are participants under the age of 18. As a result, we obtain assent from guardians or parents for all participants under the age of 18.

Line number 175 to 176

4. Result: Table 1; do you mean male partner,

 Thank you for your suggestion. We wants to say male household member who use alcohol not male partner only. 

 the way that you measured the magnitude of sexual violence is not specifically briefed in the result section (you only discussed the general magnitude of sexual violence).

 Thank you for your suggestion. We revised the suggested section in the updated manuscript.

Line number 204 to 209

 Check and take some corrections on the odds of ……the way that you discussed the factors (AOR). Table 3; avoid COR, you should include the %value of each variable, and indicate with Astrix (**) the associated factor in the final model. Thank you for your suggestion. We revised the suggested section in the updated manuscript.

 The way how did you measure the discussion about SRH should be clearly shown in the operational definition; 

the variables that you use are somehow few; in the last section of table 3 what does the average score mean (Avoid it)? Thank you for your constructive comments. We include how we measured SRH in the operational definition section.

Line number 150 to 153 

Average score mean academic performance of the student. To make it clear, we change the average score by academic performance and we avoid it in table three

5. Discussion; Line one said that These studies identify…..[correct it]……respectively (need correction) When you discuss the magnitude you should modify the grammar, the way that you discussed, and the way that you justify should be clear (It is good but it is not written scientifically). 

Discussion about factors must rewrite it as first your finding then the study which is consistent or inconsistent with your finding, reference, then possible justification (You should correct the factors section) 

Thank you for the suggestion. Based on your comment, we revised the whole discussion section in the updated manuscript including the grammar. 

Line numbers 234 to 281

 Conclusion; Is not addressed based on your pertinent finding and your objective

Check the reference some of your references do not exist in the online system. Thank you for the suggestion. We revised the conclusion section in the updated manuscript.

Line Number 282 to 289

 Reviewer 3 

1. When just starting from the title add study design and year of study. Thank you for your constructive comments. We amend all the comments on the updated manuscript as follows: "institutional-based cross-sectional study, 2022."

Line numbers 2 to 3

2. Abstract

-Remove the abbreviation from the abstract

-Merge objective and background Thank you for your constructive comments. We amended all the comments in the updated manuscript.

Line numbers 26 to 50

 - Ever had sex was significantly associated with sexual violence after being housemaids. How it could be related to lifetime sexual experience with being housemaids 

Thank you for your constructive comments. As you mentioned, having ever had sex was a significant variable, which means housemaids with a history of sex were more likely to experience sexual violence after becoming a housemaid, but does not mean that having ever had sex makes them a housemaid.

 -All variables are not self-explanatory For example: is social support low or strong?

 Thank you for your constructive comments. We amended all the comments by rename it as follows: "poor social support" in the updated manuscript. 

Line number 45 to 46. 

3. Conclusion: The magnitude of sexual violence among housemaid has been found to be high compare to previous findings. Its seems comparative study. Be fixed with your topic. Needs revision. Thank you for your comment. We accept and update the conclusion section in the updated manuscript.

Line numbers 51 to 52

4. Introduction 

Most of the introduction section deals only with violence among women in the general population rather than housemaids. Also very wide please fix with your topic. 

 Thank you for the suggestion. Based on your comment, we accept and amend all the corrections in the updated manuscript.

Line numbers 60 to 108

5. -Data collection procedure: I disagreed with the tool, it has no clear-cut point and complete or standard questionnaire. Please attach it. 

 Thank you for the suggestion. Actually, we used WHO standard violence measurement tools to assess whether housemaids were sexually violated or not. We used three yes/no questions, and if the housemaid answered yes to any of them, she was considered sexually abused. To make this more clear, we revised the operational definitions of the outcome variables as follows: “A WHO standard tool was used to assess sexual violence. Three independent Yes/No questions were asked for each housemaid. Whether or not their husband has ever physically forced them to have sex or engage in other sexual acts, they do not want to. Did your current husband/partner or any other partner ever physically force you to have sexual intercourse when you did not want to? Did you ever have sexual intercourse you did not want to have because you were afraid of what your partner or any other partner might do? Did your partner or any other partner ever force you to do something sexual that you found degrading or humiliating? If a housemaid answered yes to one of the three questions, she was considered sexually violated” 

Line number 136 to 153

 -Where pretest was conducted?

 The pretest was held in Debre Tabor Town, about 100 kilometers north of Bahir Dar. 

 - Why do you prefer P -value less than 0.25 for a multi-variable logistic regression analysis?

 Thank you for your comment. We used a p-value less than 0.25 to deduce the numbers of variables for multivariable analysis, and we believe that those variables whose p values greater than 0.25 were less likely to be statistically significant in multivariable analysis.

 - Hosmer-Lemeshow goodness test was fitted to check model fitness. 0.19. Remove the point after the word fitness. Thank you for the suggestion. Based on your comment, we amended all the corrections in the updated manuscript.

6. Result:

-Please separate the frequency and percentage of each variable with a column. 

Thank you for your suggestion. Based on your comment, we amended all the corrections in the updated manuscript. 

 -How did you categorize each variable like age, level of grade and etc?

 Thank you for the comment. We use previous research and standards to categorize each variable. For example, we categorize age as less than 20 (ages 15 to 19) and greater than or equal to 20 (>=20). Similarly, we categorize educational level based on standard categories: primary (grades 1–8), secondary (grades 9–12), and similar for the rest of the variables.

 -How did you see the difference between family support and social support? No one of the participants had strong social support. Family support is the support of families, in which family members can assist by providing medical information, making time for emotional sharing. While social support is the perception and reality that one is cared for, there is also assistance available from other people, including friends, colleagues, and acquaintances.

We used a standard Oslo 3 social support measurement scale, but no participants scored good social support on it, so we simply classified it as moderate and poor.

 Statistical analysis has not been performed appropriately. Some of the cross-tabulation (ad/bc) are written inversely and check revised correctly. Thank you for your comment. I see the statistical analysis conducted rigorously, including the cross-tabulation, and it seems fine. In some of the cross tabulations, we may have used the last categories (response variables) as reference categories that seems reversed.

 -Add P-Value for each variable in multivariable regression analysis.

……"

 Thank you for the suggestion. Based on your comment, we include p-value in the updated manuscript.

Line numbers 223

 Discussion: 

-Poor discussion compared with other studies. Revise it again.

 Thank you for the suggestion. Based on your comment, we revised the whole discussion section in the updated manuscript.

Line numbers 234 to 281

 -Hong Kong 25.38%, remove column before the reference. 

Thank you for the suggestion. Based on your comment, we amended the corrections in the updated manuscript.

 -Be consistent in using capital letters and punctuation. It has also several grammatical errors. Please show for native speakers. Thank you for your comment. To address the language issue, we consulted senior public health staff and language professors at my university. We also used online software, specifically Grammarly and scribens (which check for correct grammar and spelling), throughout the manuscript.

 - The result of your study is in line with others no need for discussion. This implies that the burden of sexual violence is more prevalent throughout the world. Despite the human right approach, everyone has the right to control their own body group of population (47). Remove it. 

Thank you for the suggestion. Based on your comment, we remove it in the updated manuscript.

 -The reason for the variation in magnitude needs further explanation. Thank you for your suggestion. We revised the entire manuscript in the updated version based on your feedback.

 -Add implication of your study Thank you for your suggestion. We include implication in the updated version based on your feedback.

 -Conclusions; add another conclusion.

 Thank you for the suggestion. We revised the conclusion section as follows: "This study identified 

---

## [Decision Letter · Decision Letter 1]

22 May 2024

PONE-D-22-33222R1Magnitude and associated factors of sexual violence among female housemaids attending night school in Bahir Dar City, North West Ethiopia: institutional based cross-sectional study, 2022.PLOS ONE

Dear Dr. Bekele,

Thank you for submitting your manuscript to PLOS ONE. After careful consideration, we feel that it has merit but does not fully meet PLOS ONE’s publication criteria as it currently stands. Therefore, we invite you to submit a revised version of the manuscript that addresses the points raised during the review process.

Please submit your revised manuscript with point-by-point responses by Jul 06 2024 11:59PM. If you will need more time than this to complete your revisions, please reply to this message or contact the journal office at plosone@plos.org. Please include the following items when submitting your revised manuscript:A rebuttal letter that responds to each point raised by the academic editor and reviewer(s). You should upload this letter as a separate file labeled 'Response to Reviewers'.A marked-up copy of your manuscript that highlights changes made to the original version. You should upload this as a separate file labeled 'Revised Manuscript with Track Changes'.An unmarked version of your revised paper without tracked changes. You should upload this as a separate file labeled 'Manuscript'.If applicable, we recommend that you deposit your laboratory protocols in protocols.io to enhance the reproducibility of your results. Protocols.io assigns your protocol its own identifier (DOI) so that it can be cited independently in the future. For instructions see: https://journals.plos.org/plosone/s/submission-guidelines#loc-laboratory-protocols. Additionally, PLOS ONE offers an option for publishing peer-reviewed Lab Protocol articles, which describe protocols hosted on protocols.io. Read more information on sharing protocols at https://plos.org/protocols?utm_medium=editorial-email&utm_source=authorletters&utm_campaign=protocols.

We look forward to receiving your revised manuscript.

Kind regards,

Philipos Petros Gile, MA

Academic Editor

PLOS ONE

Journal Requirements:

Reviewers' comments:

Reviewer's Responses to Questions

**Comments to the Author**

1. If the authors have adequately addressed your comments raised in a previous round of review and you feel that this manuscript is now acceptable for publication, you may indicate that here to bypass the “Comments to the Author” section, enter your conflict of interest statement in the “Confidential to Editor” section, and submit your "Accept" recommendation.

Reviewer #3: All comments have been addressed

Reviewer #4: (No Response)

Reviewer #5: (No Response)

2. Is the manuscript technically sound, and do the data support the conclusions?

Reviewer #3: Yes

Reviewer #4: No

Reviewer #5: Yes

3. Has the statistical analysis been performed appropriately and rigorously? 

Reviewer #3: Yes

Reviewer #4: Yes

Reviewer #5: Yes

4. Have the authors made all data underlying the findings in their manuscript fully available?

Reviewer #3: Yes

Reviewer #4: Yes

Reviewer #5: No

5. Is the manuscript presented in an intelligible fashion and written in standard English?

Reviewer #3: No

Reviewer #4: No

Reviewer #5: (No Response)

6. Review Comments to the Author

Reviewer #3: Thank you! I have seen the author's response. Still, it has several grammatical errors. Also, the response of the reviewer says “To address the language issue, we consulted senior public health staff and language professors at my university.’ It seems only one of the author provided a response. What about another authors?

Reviewer #4: 1-This manuscript needs more language edition from the beginning to the end( example: thanks, supervisors... Acknowledgments...s...also)

2- Why authors choose at this time frame (May 15 to June 20)?

3- a simple random sampling, how it could be? or how did authors made internal and external validity?

4-How was the measurement tools were adopted from literature or is it valid? how?

5-Introduction: better to start what general violence mean? background lakes idea flow from global to local

6- line 91-93: if those studies have been done what are the real significance's of your study?

7- line 101: how about Ethiopian school health program and violence?

8- background (literatures )haven't told us about school and violence?

9- line 116 and 117: how did Authors arrange for simple random sampling for both private and public schools?

10-line 124: Lacks figurative descriptions. How the private school could be represented?

11- For n, this study have done community based and your study is institutional at school how it could be a p value?

12- line 132 and 133: better to place in annex/appendix by securing privacy?

13- School mean? primary or secondary or and tertiary please see your topic and amend it.

14-line 137: operational definition need rewriting.

15-line 146: is it real validate for our context? evidence? How it could be in age difference? primary school vs secondary school age group?

16-line 150: how it could be for age lass than 12 years?

17- one MPH holder then supervisors. is it grammatically correct or make it clear?

18-line 168 and 169: how it could be for all variables mean? and line 175: please provide a tangible value.

19- line 180: how it cold be for females age less than 18 years? line 176: What innervations were done by data collectors or investigators for those who have faced violence?

20-line 181: which informed consent? your data collection sites were schools? how did you get parents/guardians?

21-line 183 and 184: what does it mean? needs grammatical correction.

22-table 1: the study were done in Bahir Dar city? what is the importance of reporting urban and rural? please see under 23- years and marital status plus how it could be only into categories? all variables in the table have confusion.

24- line 229 and 230: better to describe by percentage rather than inversion of odds ratio.

25- line 284: base line to report high? and 293: What does it mean did you have evidence( at least up to your sample size)?

Reviewer #5: Overall, the paper is well written and the objective is clear except some concerns needs to be considered:

1. Except some fragmented sentence

2. Some portions in the document needs clarification and rewritten

3. The introduction part needs to be rewritten again by keeping logical sequence

4. English language improvement is needed in the whole document.

Method Section

1. The data for the total population of Bahir Dar City is not Up to date, Please update it accordingly.

2. How do you identify the adolescents living 6 month in the city?

3. How can you justify weather some sub cities have a more administrative kebeles than others comparably, How could you make it proportional.

4. What was the K value for SRS bases household sampling

5. It would be better if you summarize by drawing the sampling tree in short and precise way?

6. Is the pretest done for this study and how was the result of Cronbach alpha value to validate the tool?

7. How do you evaluate the presence of other psychiatric condition like MDD and Schizophrenia and Epileptic conditions as it may contribute more to the suicidal attempt?

8. Have you done any crosscheck in some samples through the other group who are responsible for the data quality to minimize bias by the data collectors?

9. The data availability statement is not mentioned here?

7. PLOS authors have the option to publish the peer review history of their article (what does this mean?). If published, this will include your full peer review and any attached files.

Reviewer #3: No

Reviewer #4: No

Reviewer #5: No

---

## [Author Response · Author response to Decision Letter 1]

1 Jul 2024

Point-by-point response

We sincerely thank the reviewers for their thorough review of the manuscript and for providing constructive feedback. We believe we have thoroughly incorporated their comments into the updated version.

Reviewers Comment Revised versions

Reviewer 1. 

This manuscript needs more language editions. 

Why authors choose this time frame (May 15 to June 20)? There was no specific reason for choosing May and June, but it aligned with our work schedule. Additionally, the academic calendar in Ethiopia runs from September to June. Therefore, we collected the data before the students left for vacation. Given the total number of eligible study participants, the sample size, and the sampling technique, approximately 35 days were sufficient for collecting data from the entire sample population. 

A simple random sampling, how it could be? or how did authors made internal and external validity? This study included all sixteen schools, both private and government, that offer evening programs. Initially, a preliminary census was conducted at each school to identify students working as housemaids by recording their school names, job types, grade levels, class numbers, and sections as a sampling frame. Subsequently, participants were selected using a computer-generated method from the list of the student.

The authors aimed to maintain the internal validity of this research through the careful design of the data collection tool, ensuring appropriate measurements were used for each variable included in the study. Similarly, training was given for data collection and the process was monitored by trained supervisors. Additionally, to maintain the generalizability of the findings, the sample size was meticulously determined. 

How was the measurement tools were adopted from literature or is it valid? how?

 The data measurement tool was developed by reviewing various literature and WHO standard tools. After its development, both content and face validity were assessed by subject matter experts. Additionally, the tool was pretested to check consistency in the local context.

Introduction: better to start what general violence mean? background lakes idea flow from global to local. Thank you for the constructive comment. Based on your suggestion we include the general violence definition in the revised manuscript. 

“Violence is the intentional use of physical force or power, whether threatened or actual, against another person, which results in or has a high likelihood of resulting in injury, death, or psychological harm. While the majority of violence survivors are girls and women, boys and men can also be victims.”

line 91-93: if those studies have been done what are the real significance's of your study? Thank you for raising this issue. Although sexual violence and its determinates in Ethiopia have been studied to some extent, there remains a notable gap concerning night school students employed as housemaids. Previous studies have shown a wide range in reported prevalence rates of violence (27.8% to 72%), indicating a need for further investigation to reconcile these discrepancies. Therefore, this study aims to contribute recent evidence on sexual violence among housemaids who attend night school, aiming to inform targeted local and national interventions for this overlooked segment of population.

Line 101: how about Ethiopian school health program and violence? Thank you for the constructive comment. Ethiopia's Growth and Transformation Plan II prioritizes addressing violence as a key focus area. However, the school health programs in Ethiopia are not well functioning. Consequently, essential services related to addressing violence are unavailable in the majority of schools across the country. 

Background (literatures )haven't told us about school and violence? Thank you for the constructive comment. Based on your suggestion we revised the whole background section. 

line 116 and 117: how did Authors arrange for simple random sampling for both private and public schools? Thank you for the constructive comment. This study encompassed all sixteen schools, both private and government, that offer evening programs. Initially, a preliminary census was conducted at each school to identify students working as housemaids by recording their school names, job types, grade levels, class numbers, and sections. Each eligible respondent was then assigned a unique code for identification. Participants were selected using a simple random sampling technique, utilizing computer-generated methods.

line 124: Lacks figurative descriptions. How the private school could be represented? Thank you for your valuable feedback. We did not employ proportional allocation for private and government schools because we did not consider schools as a variable, given that there is no difference between governmental and private night schools in Ethiopia, as all night school students are supposed to pay. Additionally, no school-level difference makes them vulnerable to violence. Therefore, we employed simple random sampling, to ensure that each student had an equal chance of being represented. 

For n, this study have done community based and your study is institutional at school how it could be a p value? Thank you for your valuable feedback. Since there were few institutional-based studies conducted near our study area, we determined the sample size based on proportions derived from community-based studies. 

line 132 and 133: better to place in annex/appendix by securing privacy? Thank you for your valuable feedback. Individual identifiers such as names and household numbers/addresses were not collected to maintain ethical standards. However, for sampling purposes during the preliminary census, data such as school names, job types, grade levels, class numbers, and sections were recorded. This approach ensured that their privacy and confidentiality were not compromised.

School mean? primary or secondary or and tertiary please see your topic and amend it. Thank you for your input. In this study, “school” refers to both primary and secondary education institutes, which is the common definition in the country. Whereas tertiary education is considered as college or university. Therefore, our study only considered schools (primary or secondary).

line 137: operational definition need rewriting. Thank you for your valuable feedback. Based on your comment we revised the operational definition in the revised manuscript. 

“Social support: Social support was measured using the Oslo 3 Social Support Scale, which consists of three questions with a total possible score of 14 points. Participants scoring 3–8 were categorized as having poor social support, those scoring 9–11 were categorized as having moderate social support, and those scoring 12–14 were categorized as having good social support.”

line 146: is it real validate for our context? evidence? How it could be in age difference? primary school vs secondary school age group? Thank you for your input. We did not conduct a stratification analysis based on age and educational attainment. However, it's worth noting that most participants were around 18 years old, as indicated by the mean age of 18 years with a standard deviation of 2.28.

line 150: how it could be for age less than 12 years? Thank you for your comment. In our study, we did not observe any participants below 12 years of age.

one MPH holder then supervisors. is it grammatically correct or make it clear? Thank you for you reside the point. We corrected the error in the revised manuscript 

“ Two midwives with BSc degrees and two psychiatry nurses with BSc degrees were assigned as data collectors, with one supervisor holding a Master of Public Health (MPH)”

line 168 and 169: how it could be for all variables mean? and line 175: please provide a tangible value. Thank you for you reside the point. We included the real value in the revised manuscript.

“Descriptive statistics were computed to see the distributions of covariates.”

“The Hosmer-Lemeshow goodness test was fitted to check model fitness, which was found to be 0.65.” 

line 180: how it could be for females age less than 18 years? line 176: What innervations were done by data collectors or investigators for those who have faced violence? Thank you for your valuable comment. This study included both primary and secondary night school students, including those under 18 years old for whom parental consent was obtained. Participants who reported experiencing sexual violence were referred to community health agents for appropriate support and intervention.

line 181: which informed consent? your data collection sites were schools? how did you get parents/guardians? Thank you for your feedback. Before conducting the interviews, the data collectors made efforts to contact the parents of participants under 18 years old to obtain consent through written letters.

line 183 and 184: what does it mean? needs grammatical correction. Thank you for you reside the point. We corrected the grammatical error in the revised manuscript.

“Furthermore, all participants were informed of their right to withdraw from the study at any point.”

table 1: the study were done in Bahir Dar city? what is the importance of reporting urban and rural? Thank you for your feedback. The figure shows the place where the housemaids were born. It is not that important for current sexual violence. Therefore, we removed it from the revised manuscript. 

please see under 23- years and marital status plus how it could be only into categories? all variables in the table have confusion. Thank you for your comment. We did not have under 23 age categories. We reported two marital status categories due to zero values for other categories such as widowed and married, which are typically omitted from tables.

line 229 and 230: better to describe by percentage rather than inversion of odds ratio. Thank you for you reside the point. Based on your suggestion we revised it in the updated manuscript

line 284: base line to report high? and 293: What does it mean did you have evidence (at least up to your sample size)? Thank you for your suggestion. We revised it by including “Compared to similar literature” in the revised manuscript. 

293: What does it mean did you have evidence (at least up to your sample size)? Thank you for the feedback. During data collection, we obtained both consent and assent, including consent for publication. We omitted this section as it is not required by the journal guidelines.

Reviewer 2.

Except some fragmented sentence Thank you for the feedback. We revised the entire document. 

Some portions in the document needs clarification and rewritten 

The introduction part needs to be rewritten again by keeping logical sequence 

English language improvement is needed in the whole document. We revised the language by the senior researcher in the university and used online Grammarly. 

The data for the total population of Bahir Dar City is not Up to date, please update it accordingly. Thank you for your feedback. However, we don’t have any section which talks about Bahir Dar city's total population.

How do you identify the adolescents living 6 month in the city? Thank you for your feedback. We identify those residents through Self-report. 

How can you justify weather some sub cities have a more administrative kebeles than others comparably, How could you make it proportional. Thank you for your comment. We utilized a simple random sampling technique by identifying lists of students from the schools. As a result, we did not employ proportional allocation in our sampling approach

What was the K value for SRS bases household sampling. Thank you for your feedback. Participants were selected using a simple random sampling technique with computer-generated methods. Since the authors did not use systematic random sampling, there was no need for a sampling interval (Kth interval).

It would be better if you summarize by drawing the sampling tree in short and precise way? Thank you for your feedback. A schematic presentation of the sampling procedure doesn’t seem suitable here. As mentioned in the third response, a preliminary census was conducted, followed by coding, and finally, a simple random sampling technique was employed.

Is the pretest done for this study and how was the result of Cronbach alpha value to validate the tool? A pretest was conducted among 10% (34 participants) of the participants to check the consistency and appropriateness of the tool outside of the study area.

How do you evaluate the presence of other psychiatric condition like MDD and Schizophrenia and Epileptic conditions as it may contribute more to the suicidal attempt?

Response: this study included survivors.

 Thank you for your feedback. For this study we did not assess any psychiatric disorders as awhole.

Have you done any crosscheck in some samples through the other group who are responsible for the data quality to minimize bias by the data collectors? Thank you for your feedback. Thorough training was given to the data collectors regarding the objective the study, the data collection tool and related issues.

The data availability statement is not mentioned here? Thank you for your comment. We revised the section in the revised manuscript.

---

## [Editor Report · Decision Letter 2]

3 Jul 2024

Magnitude and associated factors of sexual violence among female housemaids attending night school in Bahir Dar City, North West Ethiopia: institutional based cross-sectional study, 2022.

PONE-D-22-33222R2

Dear Author,

We’re pleased to inform you that your manuscript has been judged scientifically suitable for publication and will be formally accepted for publication once it meets all outstanding technical requirements.

Kind regards,

Philipos Petros Gile, MA

Academic Editor

PLOS ONE
---

## [Editor Report · Acceptance letter]

16 Jul 2024

PONE-D-22-33222R2 

PLOS ONE

Dear Dr. Bekele, 

I'm pleased to inform you that your manuscript has been deemed suitable for publication in PLOS ONE. Congratulations! Your manuscript is now being handed over to our production team.

Kind regards, 

on behalf of

Dr. Philipos Petros Gile 

Academic Editor

PLOS ONE